# Inhibitory Effects of Simvastatin on IL-33-Induced MCP-1 via the Suppression of the JNK Pathway in Human Vascular Endothelial Cells

**DOI:** 10.3390/ijms241613015

**Published:** 2023-08-21

**Authors:** Katsuyuki Umebashi, Masayoshi Yamamoto, Akinori Tokito, Ku Sudou, Yoko Takenoshita, Michihisa Jougasaki

**Affiliations:** Institute for Clinical Research, National Hospital Organization Kagoshima Medical Center, Kagoshima 892-0853, Japan; umebashi.katsuyuki.gc@mail.hosp.go.jp (K.U.); yamamoto.masayoshi.bc@mail.hosp.go.jp (M.Y.); tokininn@hotmail.com (A.T.); pd3z8uos@okayama-u.ac.jp (K.S.); takenoshita.yoko.cj@mail.hosp.go.jp (Y.T.)

**Keywords:** interleukin-33, monocyte chemoattractant protein-1, cell migration, mitogen-activated protein kinase, cytokine, statins

## Abstract

An alarmin, interleukin (IL)-33 is a danger signal that causes inflammation, inducing chemotactic proteins such as monocyte chemoattractant protein (MCP)-1 in various cells. As statins have pleiotropic actions including anti-inflammatory properties, we investigated the effects of simvastatin on IL-33-induced MCP-1 expression in human umbilical vein endothelial cells (HUVECs). HUVECs were stimulated with IL-33 in the presence or absence of simvastatin. Gene expression and protein secretion of MCP-1, phosphorylation of mitogen-activated protein kinase (MAPK), nuclear translocation of phosphorylated c-Jun, and human monocyte migration were investigated. Immunocytochemical staining and Western immunoblot analysis revealed that IL-33 augmented MCP-1 protein expression in HUVECs. Real-time reverse transcription–polymerase chain reaction (RT-PCR) and enzyme-linked immunosorbent assay (ELISA) showed that IL-33 significantly increased MCP-1 mRNA and protein secretion, which were suppressed by c-jun N-terminal kinase (JNK) inhibitor SP600125 and p38 MAPK inhibitor SB203580. Simvastatin inhibited IL-33-induced MCP-1 mRNA, protein secretion, phosphorylation of JNK and c-Jun. Additionally, the IL-33-induced nuclear translocation of phosphorylated c-Jun and THP-1 monocyte migration were also blocked by simvastatin. This study demonstrated that IL-33 induces MCP-1 expression via the JNK and p38 MAPK pathways in HUVECs, and that simvastatin inhibits MCP-1 production by selectively suppressing JNK. Simvastatin may inhibit the progression of IL-33-induced inflammation via suppressing JNK to prevent MCP-1 production.

## 1. Introduction

Interleukin (IL)-33, originally identified as a nuclear factor in postcapillary high endothelial venules, is a ligand for the orphan IL-1 family receptor suppression of tumorigenicity (ST) 2 [1]. As a cytokine, IL-33 is released from various cells in response to tissue damage or mechanical strain to alert the immune system [2,3]. IL-33 acts as a danger signal with potent inflammatory properties, and therefore it is called an “alarmin”. IL-33 binds to its transmembrane receptor ST2 on the inflammatory cells, inducing Th2-associated cytokines and chemokines [1,4]. The signal transduction system induced by IL-33 includes nuclear factor-kappa B (NFκB) and mitogen-activated protein kinases (MAPKs), such as extracellular signal-regulated kinase (ERK)1/2, p38 MAPK, and c-jun N-terminal kinase (JNK) [1,5]. We recently reported that IL-33 increased the gene and protein expressions of chemokines, such as growth-regulated oncogene (GRO)-α and IL-8 in human vascular endothelial cells [6,7]. Using an antibody array assay, we also showed that IL-33 enhanced monocyte chemoattractant protein (MCP)-1 expression in human vascular endothelial cells [7]. Monocyte chemoattractant protein (MCP)-1, also called C-C motif chemokine ligand (CCL) 2, is a key member of the CC chemokine family that is involved in the pathophysiology of inflammatory diseases, and the findings that MCP-1 is a downstream molecule of IL-33 are supported by several other investigations [8,9,10,11,12,13,14,15]. Besides the property of migrating monocytes into the site of inflammation, MCP-1 also promotes migration of cancer cells [16,17,18,19,20]. In addition, accumulating evidence has demonstrated that IL-33 also promotes migration of normal cells [21,22,23] as well as cancer cells [24,25,26]. Hu et al. revealed that IL-33 and its receptor ST2 were co-expressed in decidual stromal cells, and IL-33 promoted migration of decidual stromal cells by upregulating MCP-1 via the ERK1/2 signaling pathway [27]. However, the precise role of MCP-1 on IL-33-mediated cell migration remains undefined.

Statins, 3-hydroxy-3-methylglutaryl CoA (HMG-CoA) reductase inhibitors, have been established as a primary and secondary prevention of cardiovascular events by lowering low-density lipoprotein cholesterol [28]. In addition to the cholesterol-lowering effect, statins have pleiotropic actions, including attenuation of vascular inflammation, improved endothelial cell function, stabilization of atherosclerotic plaque, decreased vascular smooth muscle cell migration and proliferation, and inhibition of platelet aggregation [29,30]. We previously reported that simvastatin suppressed IL-6-induced monocyte chemotaxis and MCP-1 expression in human vascular endothelial cells [31]. Likewise, simvastatin caused an inhibition of C-reactive protein-mediated MCP-1 secretion and migration in human primary monocytes [32]. In addition, simvastatin inhibited MCP-1 synthesis in peripheral blood mononuclear cells exposed to lipopolysaccharide and in human endothelial cells exposed to IL-1β [33]. However, the specific mechanisms by which statins suppress MCP-1 and cell migration are not fully understood.

In the present study, we hypothesized that statins could suppress IL-33-induced monocyte chemotaxis through inhibiting MCP-1 expression. Thus, this study was designed to investigate the effects of simvastatin on IL-33-induced monocyte chemotaxis and MCP-1 expression in human umbilical vein endothelial cells (HUVECs), specifically focusing on the signal transduction system of MAPK pathways.

## 2. Results

### 2.1. Immunocytochemical Staining for MCP-1

Immunocytochemical staining showed faint positive immunoreactivities for MCP-1 in untreated HUVECs, and the intensity of immunoreactivity for MCP-1 in HUVECs was enhanced by the treatment with 10^−9^ mol/L of IL-33 for 24 h (Figure 1A). In addition, the cells treated with normal IgG, instead of the primary antibody against MCP-1, demonstrated no immunoreactivity for MCP-1. Semiquantitative analysis showed a significant increase in the intensity of immunoreactivity for MCP-1 in IL-33-stimulated cells compared with the untreated cells (Figure 1B).

### 2.2. Western Immunoblot Analysis for MCP-1

Immunocytochemical findings were supported by Western immunoblot analysis of the cell lysates using anti-MCP-1 antibody (Figure 2A). As shown in Figure 2B, treatment with 10^−9^ mol/L of IL-33 resulted in a significant increase in MCP-1 protein expression in HUVECs.

### 2.3. IL-33-Induced Gene Expression and Protein Secretion of MCP-1 in HUVECs

Real-time reverse transcription–polymerase chain reaction (RT-PCR) demonstrated that treatment with IL-33 (10^−12^ to 10^−8^ mol/L) resulted in an increase in MCP-1 mRNA in a dose-dependent manner with a statistical significance at 10^−9^ and 10^−8^ mol/L of IL-33 (Figure 3A). As shown in Figure 3B, IL-33 at the dose of 10^−9^ mol/L significantly increased MCP-1 mRNA between 4 and 24 h, peaking at 8 h after stimulation with IL-33. An enzyme-linked immunosorbent assay (ELISA) showed that IL-33 increased MCP-1 protein secretion from HUVECs in a dose-dependent manner with a significant increase at the doses over 10^−11^ mol/L (Figure 3C), and in a time-dependent manner with a significant increase between 8 and 24 h of IL-33 treatment (Figure 3D).

### 2.4. IL-33-Induced Phosphorylation of JNK, p38 MAPK, and ERK1/2 in HUVECs

HUVECs were exposed to IL-33 for different time periods (5–120 min), and their protein extracts were examined by Western immunoblot analysis. IL-33 induced the phosphorylation of JNK (Figure 4A), peaking at 30 min and declining at 60 min. In addition, IL-33 induced the phosphorylation of p38 MAPK (Figure 4B) and ERK1/2 (Figure 4C), peaking between 15 min and 30 min and declining at 60 min.

### 2.5. Effects of Pharmacological Inhibitors of MAPK Signaling Pathways on Gene Expression and Protein Secretion of MCP-1 in HUVECs

Involvement of the MAPK pathway in the IL-33-induced gene expression and protein secretion of MCP-1 was examined using pharmacological inhibitors of MAPK, such as SP600125 (JNK inhibitor), SB203580 (p38 MAPK inhibitor), and PD98059 (ERK1/2 inhibitor). HUVECs were pretreated with these pharmacological inhibitors for 2 h, followed by stimulation with IL-33 (10^−9^ mol/L) for 8 h to measure MCP-1 mRNA expression, and for 24 h to examine MCP-1 protein secretion from HUVECs. Real-time RT-PCR demonstrated that the IL-33-induced increase in MCP-1 mRNA expression was significantly suppressed by the pretreatment with SP600125 and SB203580 in HUVECs (Figure 5A). As shown in Figure 5B, the IL-33-induced increase in MCP-1 protein secretion from HUVECs was also significantly attenuated by the pretreatment with SP600125 and SB203580. In contrast, PD98059 had no effect on the IL-33-induced upregulation of gene expression or protein secretion of MCP-1 in HUVECs.

### 2.6. Effects of Simvastatin on Cell Viability

MTT assay was used to examine cytotoxicity of various doses of simvastatin to the cultured HUVECs. No significant changes in cell viability were observed in HUVECs treated with simvastatin at a dose of less than 10 μmol/L (Figure 6). However, 100 μmol/L of simvastatin significantly decreased cell viability, and therefore this dose was not used in the present study.

### 2.7. Effects of Simvastatin on IL-33-Induced Gene Expression and Protein Secretion of MCP-1 in HUVECs

To investigate the effects of simvastatin on IL-33-induced gene expression and protein secretion of MCP-1, HUVECs were pretreated with various concentrations of simvastatin (0.1, 1, 10 μmol/L), followed by stimulation with IL-33 (10^−9^ mol/L) for 8 h to examine MCP-1 mRNA expression, and for 24 h to measure MCP-1 protein secretion from HUVECs. The IL-33-induced increase in MCP-1 mRNA expression was significantly inhibited by 10 μmol/L of simvastatin (Figure 7A). As shown in Figure 7B, the IL-33-induced MCP-1 protein secretion from HUVECs for 24 h was also significantly suppressed by 10 μmol/L of simvastatin.

### 2.8. Effects of Simvastatin on IL-33-Stimulated Phosphorylation of JNK and p38 MAPK in HUVECs

To evaluate whether simvastatin suppresses JNK and p38 MAPK activity, the phosphorylation of JNK and p38 MAPK was examined by Western immunoblot analysis. HUVECs were pretreated with various concentrations of simvastatin and then incubated with IL-33 (10^−9^ mol/L) for 15 min. IL-33-stimulated JNK phosphorylation was significantly inhibited by simvastatin at the dose of 10 μmol/L (Figure 8A). However, simvastatin had no effect on the phosphorylation of p38 MAPK (Figure 8B).

### 2.9. Effects of Simvastatin on IL-33-Stimulated Phosphorylation of c-Jun in HUVECs

To determine the downstream signaling pathway of JNK, phosphorylation of c-Jun was assessed by Western immunoblot analysis. IL-33 induced the phosphorylation of c-Jun, peaking at 30 min and declining at 120 min (Figure 8C). The effects of simvastatin on the phosphorylation of c-Jun were examined by the pretreatment of HUVECs with various concentrations of simvastatin, followed by stimulation with IL-33 (10^−9^ mol/L) for 30 min. IL-33-stimulated c-Jun phosphorylation was significantly suppressed by the addition of simvastatin at the dose of 10 μmol/L (Figure 8D).

### 2.10. Immunofluorescence Staining

HUVECs were pre-incubated with simvastatin followed by incubation with IL-33 for 30 min, and immunofluorescence staining was performed to examine whether simvastatin affects the translocation of c-Jun to the nucleus by inhibiting IL-33-induced c-Jun phosphorylation. The immunofluorescence signal of phospho-c-Jun was localized in the nuclei of HUVECs after incubation with IL-33 for 30 min compared with the untreated control cells (Figure 9A). The phospho-c-Jun activation by IL-33 was inhibited by simvastatin at the dose of 10 μmol/L. The addition of mevalonate reversed the phosphorylation. The percentage of phospho-c-Jun-positive cell nuclei was significantly increased by IL-33 stimulation, and simvastatin treatment reduced the number of phospho-c-Jun-positive cells, which was reversed by the addition of mevalonate (Figure 9B).

### 2.11. Simvastatin Reduces THP-1 Monocyte Chemotaxis Enhanced by IL-33-Induced MCP-1

The culture medium from IL-33-treated HUVECs increased migration of THP-1 cells compared with that from the untreated cells. Pre-incubation of the culture medium with goat anti-human MCP-1 polyclonal antibody inhibited IL-33-enhanced THP-1 migration; however, goat IgG had no effect on THP-1 monocyte migration, indicating that IL-33-induced THP-1 monocyte migration was, at least in part, due to the chemotactic actions of MCP-1 (Figure 10). Pretreatment of HUVECs with simvastatin at a concentration of 10 μmol/L suppressed IL-33-enhanced THP-1 monocyte migration, and the addition of mevalonate reversed the THP-1 monocyte migration (Figure 10).

## 3. Discussion

Chemokines are small-molecule inflammatory proteins that are divided into four canonical subclasses according to the position of N-terminal cysteine residues: C, CC, CXC, and CX3C chemokines [34]. MCP-1, also known as CCL2, plays a central role in the pathogenesis of several different disease processes, including vascular permeability and attraction of immune cells during metastasis, various neurological disorders, autoimmune disease, obesity, and atherosclerosis [35]. We previously reported that IL-33 enhanced MCP-1 protein expression in HUVECs by the method of antibody array assay [7]. Previous studies also demonstrated that IL-33 increased MCP-1 expression in various cells, including human and mouse mast cells [8,10,12,14], human corneal epithelial cells [11], human vascular endothelial cells [9,15], and human cancer cells [13,26]. In the present study, immunocytochemical examination and Western immunoblot assay revealed that IL-33 increased MCP-1 protein expression in HUVECs. In addition, real-time PCR and ELISA demonstrated that IL-33 increased gene expression and protein secretion of MCP-1 in a dose- and time-dependent manner in HUVECs. The biological roles of IL-33-induced MCP-1 upregulation in various cells need further investigation.

Extracellular IL-33 binds transmembrane receptor ST2 and causes ST2-dependent signaling pathways including MAPK pathways, such as ERK1/2, p38 MAPK, and JNK. Signal transduction pathways involved in IL-33-induced MCP-1 activation were investigated by previous studies. Yagami et al. demonstrated that the IL-33-mediated synthesis of MCP-1 were dramatically and dose-dependently reduced by the addition of p38 MAPK inhibitor SB202190 but not by ERK inhibitor PD98059, indicating that p38 MAPK is required for the IL-33-mediated increase in MCP-1 in human vascular endothelial cells [15]. Another study showed that p38 MAPK inhibition with SB203580 suppressed IL-33-induced MCP-1 secretion without inhibiting MCP-1 gene expression in human skin mast cells, suggesting post-transcriptional involvement [10]. IL-33-induced MCP-1 gene expression was significantly inhibited by either MEK inhibitor U0126 or p38 MAPK inhibitor SB203580, suggesting that both ERK1/2 and p38 MAPK pathways are involved in the IL-33-induced upregulation of MCP-1 in bone marrow-derived mast cells [14]. In the present study, upregulation of MCP-1 mRNA and protein secretion induced by IL-33 in HUVECs was significantly suppressed by p38 MAPK inhibitor SB203580 and JNK inhibitor SP600124, suggesting that IL-33-induced MCP-1 upregulation involves both p38 MAPK and JNK pathways in HUVECs. These findings were supported by the previous investigation that stimulation of human mast cells with IL-33 significantly increased MCP-1 secretion via p38 MAPK and JNK pathways, with a higher concentration of JNK inhibitor required to inhibit MCP-1 release in human mast cells [8]. Further studies are required to explore the role of the MAPK pathways in the biological actions of IL-33 in various types of cells.

Cell migration is essential for proper immune response, wound repair, and tissue homeostasis, while aberrant cell migration is found in various pathological conditions [36]. MCP-1 plays an important role in migration of not only monocytes/macrophages but also cancer cells. MCP-1 induces cancer cell abscission, migration, and invasion in both autocrine and paracrine manners, playing a pivotal role in tumor metastasis [19,20]. In addition to the cell-migrating properties of MCP-1, accumulating evidence has revealed that IL-33 also promotes migration of various cells. IL-33 increased the migration of human endothelial cells, playing an important role in angiogenesis [21] and lymphangiogenesis [22]. IL-33 also stimulated the migration and invasion of human gastric cancer cells [24], human lung cancer cells [25], and human esophageal cancer cells [26]. Lin et al. demonstrated that IL-33 enhanced cell migration and invasion via inducing the epithelial-to-mesenchymal transition by JNK activation in human glioma cells [37]. Tjota et al. [38] demonstrated that IL-33 promoted the expression of multiple chemokines including MCP-1 and that exogenous recombinant IL-33 migrated monocytes to the lung interstitium. The significant roles of IL-33-induced MCP-1 were examined by other investigations. Hu et al. reported that IL-33 and its receptor ST2 were co-expressed in decidual stromal cells, and IL-33 stimulated the activation of NFκB and ERK1/2 to increase the expression of MCP-1, thereby promoting the migration and invasion of decidual stromal cells [27]. IL-33 promoted cancer cell migration and invasion via inducing epithelial-to-mesenchymal transition by the activation of MCP-1 in esophageal squamous cell carcinoma [26]. Although these findings raise the possibility that IL-33 might promote cell migration via activating MCP-1 in various cells, the precise roles of MCP-1 on IL-33-promoted normal or cancer cell migration remain undefined. In the current study, we confirmed that IL-33 promoted THP-1 migration, which was significantly suppressed by anti-MCP-1 antibody, suggesting that MCP-1 induced by IL-33 plays an important role in the migration of monocytes. Taken together, these results suggest that IL-33 is involved in the pathophysiology of cell migration due to upregulation of MCP-1 in vascular endothelial cells.

Statins, HMG-CoA reductase inhibitors, are cholesterol-lowering drugs that are widely prescribed in the treatment of cardiovascular diseases. Statins exert numerous pleiotropic effects including anti-inflammatory actions [30]. Indeed, we have already demonstrated that simvastatin reduces IL-6-induced monocyte chemotaxis and MCP-1 expression in human vascular endothelial cells by inhibiting the Janus kinase-signal transducers and activators of transcription (JAK-STAT) pathway [31]. The effects of statins on IL-33-mediated inflammation have been reported in the literature. Montecucco et al. demonstrated that simvastatin inhibited C-reactive protein-induced MCP-1 secretion and monocyte migration through the inhibition of the ERK1/2 signaling pathway [32]. Romano et al. reported that simvastatin also caused a dose-dependent inhibition of MCP-1 production in peripheral blood mononuclear cells exposed to lipopolysaccharide and in human endothelial cells exposed to IL-1β [33]. The present study revealed that simvastatin suppressed MCP-1 gene expression and protein secretion induced by IL-33 in HUVECs. Furthermore, it is interesting that simvastatin only suppressed the IL-33-mediated phospho-JNK pathway but not the p38 MAPK pathway. In addition, simvastatin suppressed the phosphorylation of c-Jun and its translocation to the nucleus in HUVECs in the present study. These findings suggest that simvastatin may act as a JNK inhibitor in the treatment of inflammation.

The simvastatin concentrations used in the present study are higher than the therapeutic plasma concentrations of simvastatin in clinical situations with humans. The present study demonstrated that 10 μmol/L of simvastatin inhibited the gene expression and protein secretion of MCP-1 as well as the phosphorylation of JNK and c-Jun in HUVECs. In pharmacokinetic studies, Lilja et al. [39] reported that the maximal plasma concentrations of simvastatin in human subjects receiving 40 mg of simvastatin daily were almost 5–30 ng/mL (0.01–0.07 μmol/L). However, previous in vitro studies have also reported that simvastatin is used with similar concentrations as those of the current study in monocytes, osteoblasts, and vascular endothelial cells [31,40,41]. In addition, the duration of exposure to statins in HUVECs should be considered in the in vitro cell culture experiments. Despite low levels of simvastatin in plasma, cells are constantly exposed to simvastatin and may be accumulated intracellularly. As the time of exposure of cells to simvastatin is very short, usually only for an hour, any significant inhibition of the MCP-1 gene and protein expression induced by IL-33 might require higher concentrations of simvastatin in the in vitro experiments. The clinical relevance of simvastatin concentrations in the in vitro studies needs further investigation.

## 4. Materials and Methods

### 4.1. Regents

Recombinant human IL-33 was purchased from Pepro Tech (Rocky Hill, NJ, USA). The mouse monoclonal anti-human MCP-1 antibody was from Santa Cruz Biotechnology (Heidelberg, Germany). The rabbit polyclonal antibodies for JNK, phospho-JNK (Thr183/Tyr185), p38, phospho-p38 (Thr180/Tyr182), ERK1/2, phospho-ERK1/2 (p42/44 MAPK), c-Jun, and phospho-c-Jun (Ser73) were obtained from Cell Signaling Technology (Beverly, MA, USA). SP600125 (JNK inhibitor) was purchased from BIOMOL (Plymouth Meeting, PA, USA). Simvastatin, PD98059 (ERK1/2 inhibitor), and SB203580 (p38 MAPK inhibitor) were purchased from FUJIFILM Wako Pure Chemical (Osaka, Japan). Mevalonate was obtained from Sigma (St Louis, MO, USA).

### 4.2. Cell Culture of HUVECs

HUVECs were purchased from Kurabo (Osaka, Japan) and seeded in plastic plates precoated with type I collagen (Asahi Techno Glass, Nagoya, Japan) and were maintained in endothelial cell growth medium (Promo cell, Heidelberg, Germany) supplemented with 0.5 μg/mL fungizone, 0.25 μg/mL amphotericin B, 100 μg/mL streptomycin, and 100 U/mL penicillin (Life Technologies, Carlsbad, CA, USA).

### 4.3. Immunocytochemical Staining

HUVECs incubated on a Biocoat slide glass (BD Biosciences, San Jose, CA, USA) were fixed with 4% buffered paraformaldehyde (FUJIFILM Wako Pure Chemical, Osaka, Japan) for 20 min. The indirect immunoperoxidase method was used for the immunocytochemical analysis as described previously [42]. The primary antibody against MCP-1 was used at 100-fold dilution. The specificity of the staining was confirmed by substitution of the normal mouse IgG for the primary antibody. The intensity of staining was semiquantitatively evaluated by two independent examiners. Grades for the staining intensity ranged from 0 to 3, with 0 indicating no staining; 1, weak staining; 2, moderate staining; and 3, strong staining.

### 4.4. Western Immunoblot Analysis

Western immunoblot analysis was performed as described previously with some modifications [42,43]. In brief, HUVECs were lysed with ice-cold cell lysis buffer together with phenylmethylsulphonyl fluoride and protease inhibitor cocktail. The harvested cells were resuspended in sodium dodecyl sulfate sample buffer and dithiothreitol, sonicated, and boiled for 5 min. They were separated by 4–12% NuPAGE Bis-Tris gels (Life Technologies, Carlsbad, CA, USA) and transferred to a polyvinylidene difluoride membrane by electroblotting for 2 h. The membrane was soaked in 5% nonfat dry milk blocking buffer. The membrane was then incubated with the primary antibody overnight at 4 °C at concentrations as suggested by the manufacturer, followed by incubation with horseradish peroxidase-conjugated secondary antibody (Cell Signaling Technology, Beverly, MA, USA) for 1 h. The protein bands were visualized by ECL prime (GE Healthcare, Buckinghamshire, UK), and the intensities of the blots were analyzed by a ChemiDoc Touch Imaging System (Bio-Rad, Hercules, CA, USA).

### 4.5. Total RNA Extraction and Real-Time RT-PCR

A Pure Link RNA Mini kit (Invitrogen, Carlsbad, CA, USA) was used for the extraction of total RNA from HUVECs, and cDNA was synthesized with a Superscript VILO cDNA Synthesis kit (Invitrogen, Carlsbad, CA, USA). Real-time PCR was performed using Power SYBR Green PCR Master Mix (Applied Biosystems, Warrington, UK) on a CFX connect thermal cycler (Bio-Rad, Hercules, CA, USA). The value of each cDNA was calculated using the ΔΔCq method and normalized to the value of the housekeeping gene glyceraldehyde-3-phosphate dehydrogenase (GAPDH).

Oligonucleotide PCR primers targeting MCP-1 mRNA were designed according to a previous report [44], and the specificity of the primers was confirmed by BLAST search and melting curve analysis. The primer sequences are shown in Table 1.

The reaction conditions were as follows: activation step at 95 °C for 10 min, followed by 40 cycles of denaturation at 95 °C for 15 s and annealing/extension at 60 °C for 1 min.

### 4.6. Enzyme-Linked Immunosorbent Assay (ELISA)

Concentrations of MCP-1 in the culture medium were determined by using a human MCP-1 ELISA kit (R&D Systems, Minneapolis, MN, USA) according to the manufacturer’s protocol. The optical densities of samples and standards were measured spectrophotometrically with an iMark microplate reader (BIORAD, Hercules, CA, USA). MCP-1 concentrations were determined by comparison of the optical density results with the standard curve.

### 4.7. Cell Viability

Cell viability was determined based on the MTT assay (Roche, Mannheim, Germany). HUVECs were treated with 0.01, 0.1, 1, 10, or 100 μmol/L simvastatin for 24 h, after which 0.5 mg/mL MTT solution was added to the culture medium, and then incubated for 4 h. After adding dimethyl sulfoxide to the cells, the absorbance at 570 nm was measured with an iMark microplate reader (BIORAD, Hercules, CA, USA). The survival rates of the simvastatin-treated cells were compared with those of the control untreated cells.

### 4.8. Immunofluorescence Staining

HUVECs plated on a BioCoat slide glass (BD biosciences, San Jose, CA, USA) were fixed with 4% paraformaldehyde and permeabilized with 0.1% Triton X-100. They were blocked with normal horse serum for 30 min and incubated with the rabbit phospho-c-Jun antibody at 800-fold dilution overnight. Then, they were washed and incubated with anti-rabbit IgG-Alexa (Cell Signaling Technology, Beverly, MA, USA) at 250-fold dilution for 1 h, and the nuclei were counterstained with Hoechst 33342 (Invitrogen, Carlsbad, CA, USA) for 5 min. Images were analyzed by fluorescence microscope (Olympus, Tokyo, Japan). The percentage of phospho-c-Jun-positive cell nuclei was evaluated in five fields of each slide.

### 4.9. Chemotaxis Assay

A microchemotaxis chamber with polyvinylpyrrolidone-free polycarbonate filter (5 μm pore size) was used for the chemotaxis assay. THP-1 monocytic cells (8 × 10^6^ cells/mL) were plated in the upper wells of ChemoTx microplates (Neuro Probe Inc. Gaithersburg, MD, USA). The culture medium derived from untreated or IL-33-treated cells was added to the lower wells. The number of THP-1 cells migrated to the lower chamber was counted by a hemocytometer. The culture medium from the untreated cells supplemented with recombinant human MCP-1 at 100 nmol/L (PeproTech, Rocky Hill, NT, USA) served as a positive control. Normal goat IgG (R&D Systems, Minneapolis, MN, USA) was used as a negative control. To evaluate MCP-1 specific chemotaxis, anti-human MCP-1 polyclonal antibody (R&D Systems, Minneapolis, MN, USA) was added at 80 μg/mL to neutralize the secreted MCP-1.

### 4.10. Statistical Analysis

Data are shown as mean ± SD. Each data point represents the average of three to six independent experiments. Statistical significance of the data was assessed by one-way ANOVA with the Tukey–Kramer’s post hoc test. *p* value < 0.05 was considered statistically significant.

## 5. Conclusions

In conclusion, the present study demonstrated that IL-33 induces gene expression and protein secretion of MCP-1 through the activation of the JNK and p38 MAPK pathways in human vascular endothelial cells. Furthermore, simvastatin suppressed MCP-1 production by selectively inhibiting the JNK pathway. We speculate that the increase in local and circulating IL-33 levels in patients with inflammatory disease would stimulate the vascular endothelial cells to enhance MCP-1 production via the JNK and p38 MAPK pathways. Simvastatin could reduce IL-33-mediated MCP-1 production by inhibiting the JNK pathway and suppressing the recruitment of monocytes into the inflammatory lesions. These findings indicate that simvastatin may be potentially utilized as a novel therapeutic strategy for IL-33-associated inflammation.

## Figures and Tables

**Figure 1 ijms-24-13015-f001:**
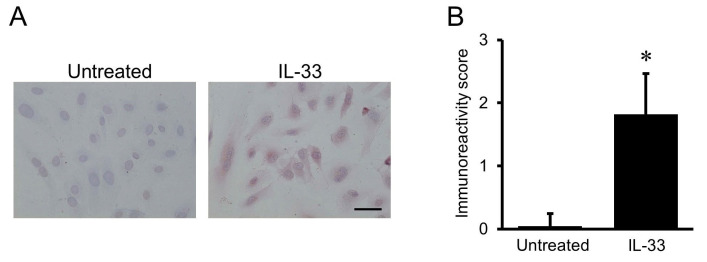
Interleukin (IL)-33-stimulated monocyte chemoattractant protein (MCP)-1 protein expression in human umbilical vein endothelial cells (HUVECs). (**A**) Representative immunocytochemical staining showing the localization of MCP-1 in HUVECs with or without exposure to 10^−9^ mol/L of IL-33 for 24 h. Intensity of immunoreactivity for MCP-1 was increased in HUVECs treated with IL-33 compared with the untreated cells. Original magnification: ×400. Scale bar = 50 μm. (**B**) Semiquantitative analysis of staining intensity of immunoreactivity for MCP-1. * *p* < 0.05 vs. untreated cells.

**Figure 2 ijms-24-13015-f002:**
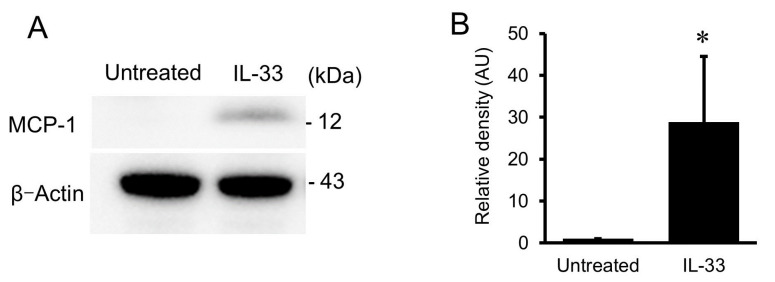
(**A**) Western immunoblot analysis of the whole-cell lysates using anti-MCP-1 antibody in HUVECs with or without exposure to 10^−9^ mol/L of IL-33 for 24 h. (**B**) Bars represent densitometric data of each expression signal after normalization to β-actin and relative to the untreated cells. MCP-1 protein expression was increased in HUVECs treated with IL-33 compared with the untreated cells. * *p* < 0.05 vs. untreated cells.

**Figure 3 ijms-24-13015-f003:**
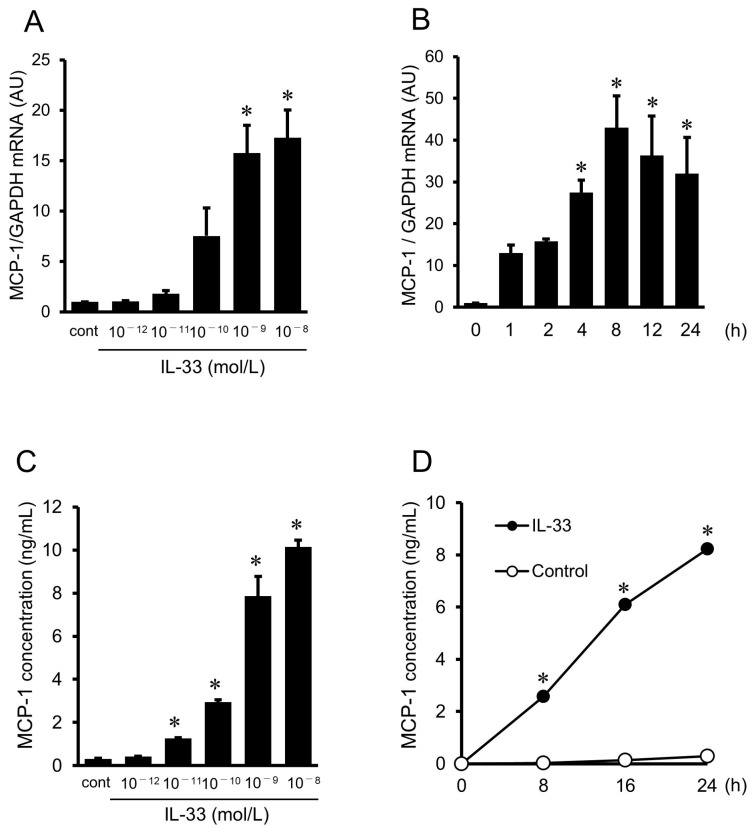
IL-33-stimulated gene expression and protein secretion of MCP-1 in HUVECs. (**A**) MCP-1 mRNA expression in HUVECs after treatment with the indicated concentrations of IL-33 for 8 h (*n* = 3), as evaluated by real-time reverse transcription–polymerase chain reaction (RT-PCR). (**B**) Time course of MCP-1 mRNA after treatment with 10^−9^ mol/L of IL-33 (*n* = 3), as evaluated by real-time RT-PCR. Bars represent MCP-1 mRNA after normalization to glyceraldehyde-3-phosphate dehydrogenase (GAPDH) mRNA and relative to the untreated control (cont) in (**A**) and 0 h in (**B**). (**C**) MCP-1 concentrations in the supernatant after treatment with the indicated concentrations of IL-33 for 24 h (*n* = 6), as analyzed by enzyme-linked immunosorbent assay (ELISA). Bars represent MCP-1 protein secretion per 10^5^ cells. (**D**) Time course of MCP-1 concentrations in the supernatant after treatment with 10^−9^ mol/L of IL-33 (closed circles, *n* = 6), as analyzed by ELISA. Spontaneous secretion of MCP-1 without IL-33 treatment is shown in open circles (*n* = 6), as analyzed by ELISA. * *p* < 0.05 vs. cont in (**A**,**C**), vs. 0 h in (**B**), and vs. each control at the same time in (**D**).

**Figure 4 ijms-24-13015-f004:**
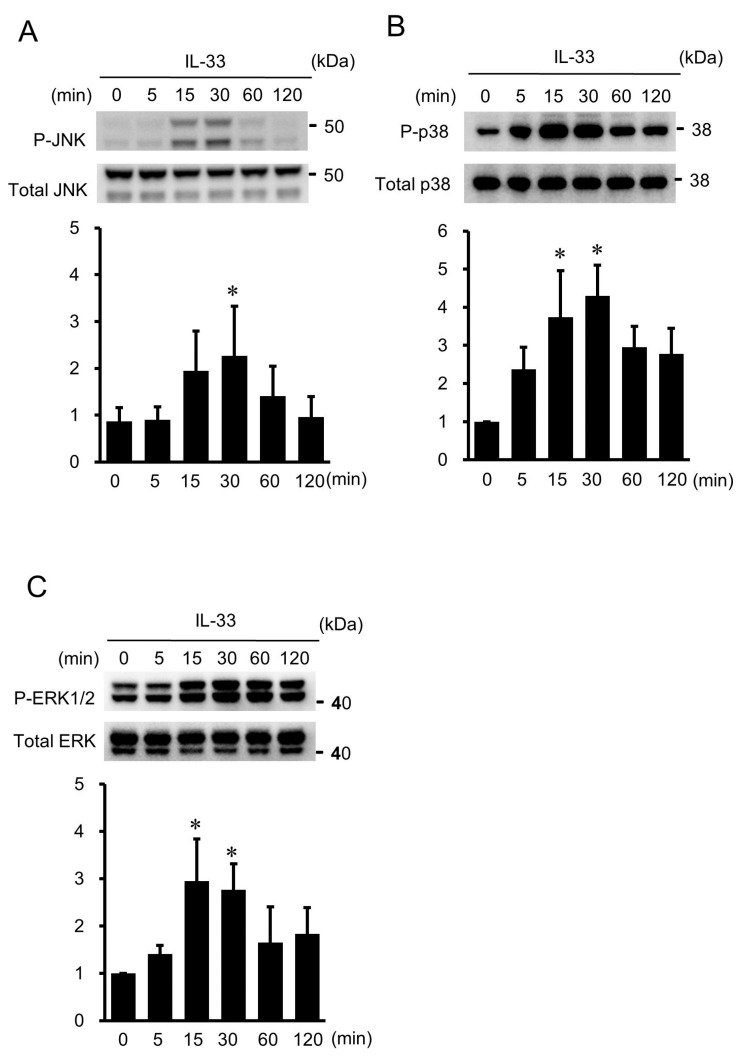
IL-33-activated mitogen-activated protein kinase (MAPK) pathway in HUVECs. (**A**–**C**) Western immunoblot analysis showed that IL-33 stimulated the phosphorylation of c-jun N-terminal kinase (JNK) (**A**), p38 MAPK (**B**), and extracellular signal-regulated kinase (ERK) 1/2 (**C**). HUVECs were treated with 10^−9^ mol/L of IL-33 for 5, 15, 30, 60, and 120 min. Bars represent results from densitometric analyses of each phosphorylation signal after normalization to total protein and relative to the untreated control (0 min). Blots are representative of three independent experiments. * *p* < 0.05 vs. 0 min.

**Figure 5 ijms-24-13015-f005:**
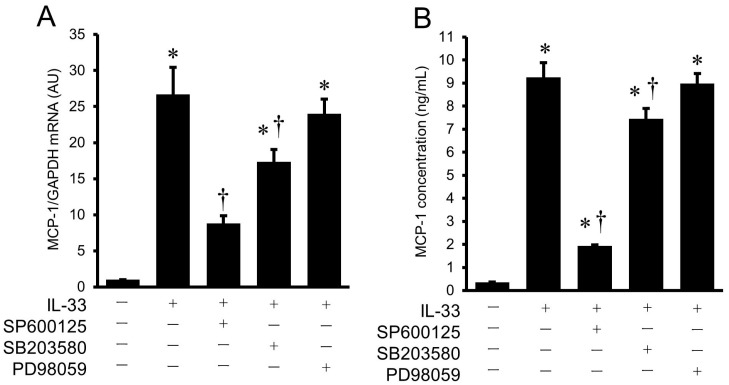
Effects of pharmacological inhibitors of MAPK pathway on IL-33-induced gene expression and protein secretion of MCP-1 in HUVECs. (**A**,**B**) HUVECs were pre-incubated with SP600125 (30 μmol/L), SB203580 (10 μmol/L), and PD98059 (30 μmol/L) for 2 h, followed by stimulation with IL-33 (10^−9^ mol/L) for 8 h ((**A**), MCP-1 mRNA) or with IL-33 (10^−9^ mol/L) for 24 h ((**B**), MCP-1 secretion). MCP-1 mRNA was evaluated by real-time RT-PCR ((**A**), *n* = 3), and MCP-1 concentration was examined by ELISA ((**B**), *n* = 6). * *p* < 0.05 vs. untreated control. † *p* < 0.05 vs. IL-33.

**Figure 6 ijms-24-13015-f006:**
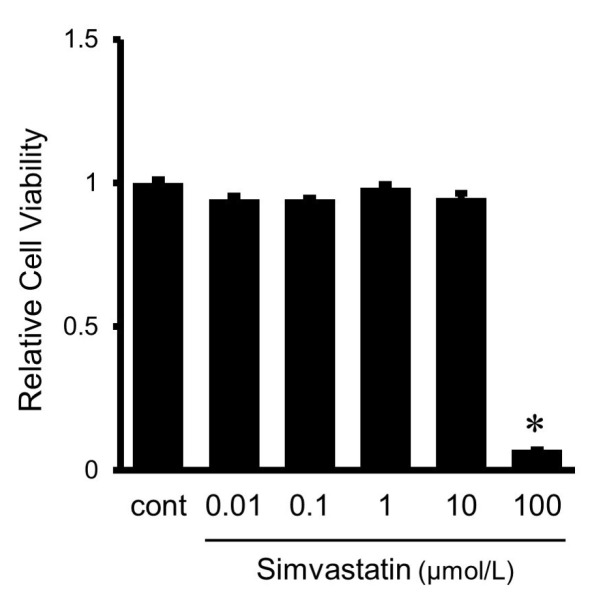
Effects of simvastatin on cell viability. HUVECs were treated with different concentrations of simvastatin for 24 h. Cell viability was measured by MTT assay. The results are expressed as percentage of the untreated control, and each value represents five independent experiments (*n* = 5). * *p* < 0.05 vs. untreated control (cont).

**Figure 7 ijms-24-13015-f007:**
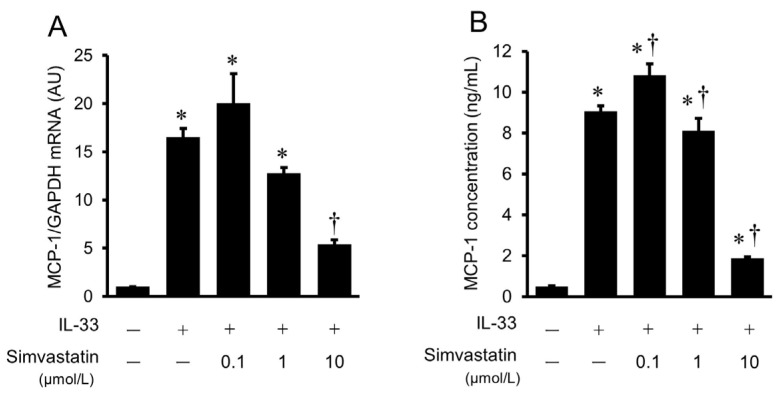
Effects of simvastatin on gene expression and protein secretion MCP-1 in HUVECs. IL-33-induced MCP-1 gene expression (**A**) and MCP-1 protein secretion (**B**) were suppressed by simvastatin. HUVECs were treated with IL-33 (10^−9^ mol/L) for 8 h (**A**) or 24 h (**B**) with or without pretreatment with simvastatin (0.1 to 10 μmol/L). Bars represent MCP-1 mRNA after normalization to GAPDH mRNA and relative to the untreated control in (**A**). Bars represent MCP-1 protein secretion per 10^5^ cells in (**B**). * *p* < 0.05 vs. untreated control. † *p* < 0.05 vs. IL-33.

**Figure 8 ijms-24-13015-f008:**
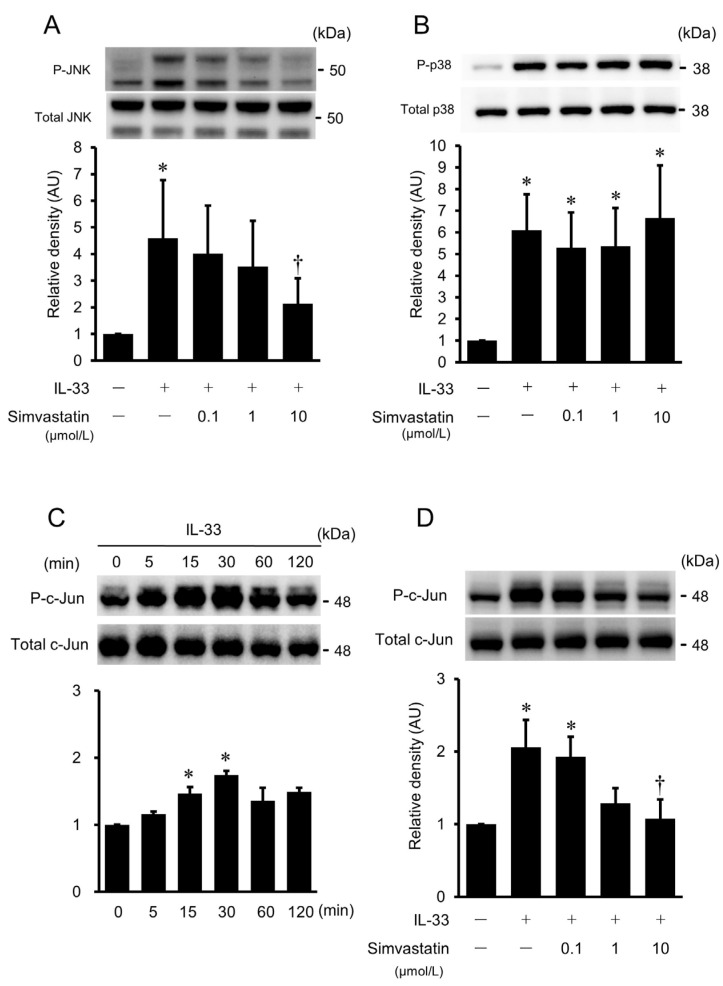
Effects of simvastatin on IL-33-induced phosphorylation of JNK, p38 MAPK, and c-Jun. (**A**,**B**) Simvastatin dose-dependently suppressed phosphorylation of JNK (**A**) but did not inhibit phosphorylation of p38 MAPK (**B**). HUVECs were pretreated with simvastatin and then incubated with IL-33 (10^−9^ mol/L) for 15 min. Bars represent results from densitometric analyses of each phosphorylation signal after normalization to total protein and relative to the untreated control. Blots are representative of three independent experiments. * *p* < 0.05 vs. untreated control. † *p* < 0.05 vs. IL-33. (**C**) Time course of IL-33-induced phosphorylation of c-Jun as evaluated by Western immunoblot analysis. HUVECs were treated with IL-33 (10^−9^ mol/L) for the indicated time periods. Bars represent results from densitometric analyses of each phosphorylation signal after normalization to total protein and relative to the untreated control (0 min). Blots are representative of three independent experiments. * *p* < 0.05 vs. 0 min. † *p* < 0.05 vs. IL-33. (**D**) Simvastatin dose-dependently suppressed phosphorylation of c-Jun. HUVECs were pretreated with simvastatin and then incubated with IL-33 (10^−9^ mol/L) for 30 min. Bars represent results from densitometric analyses of each phosphorylation signal after normalization to total protein and relative to the untreated control. Blots are representative of three independent experiments. * *p* < 0.05 vs. untreated control. † *p* < 0.05 vs. IL-33.

**Figure 9 ijms-24-13015-f009:**
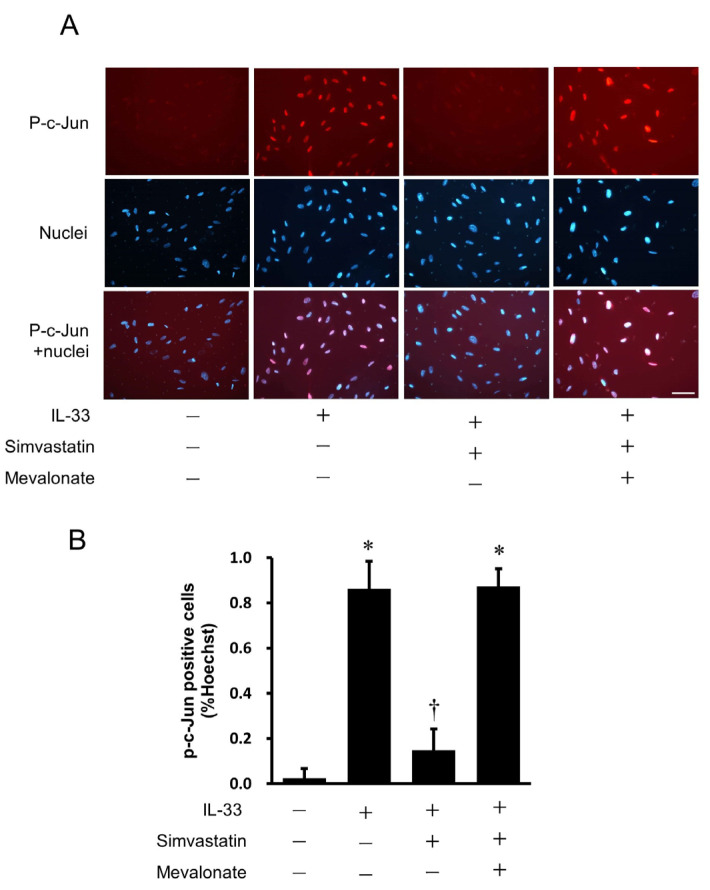
Effects of simvastatin on IL-33-induced translocation of phospho-c-Jun to the nucleus as determined by immunofluorescence staining. HUVECs were pretreated with simvastatin (10 μmol/L) or simvastatin plus mevalonate (100 μmol/L), followed by additional incubation with IL-33 (10^−9^ mol/L) for 30 min. (**A**) Representative immunofluorescence images showing the localization of phospho-c-Jun in HUVECs. Red staining indicates the specific Alexa staining for phospho-c-Jun, and blue staining indicates the nuclei (Hoechst 33342). Original magnification: ×400. Scale bar = 50 μm. (**B**) Percentages of phospho-c-Jun-positive cells relative to total cell numbers. * *p* < 0.05 vs. untreated control. † *p* < 0.05 vs. IL-33.

**Figure 10 ijms-24-13015-f010:**
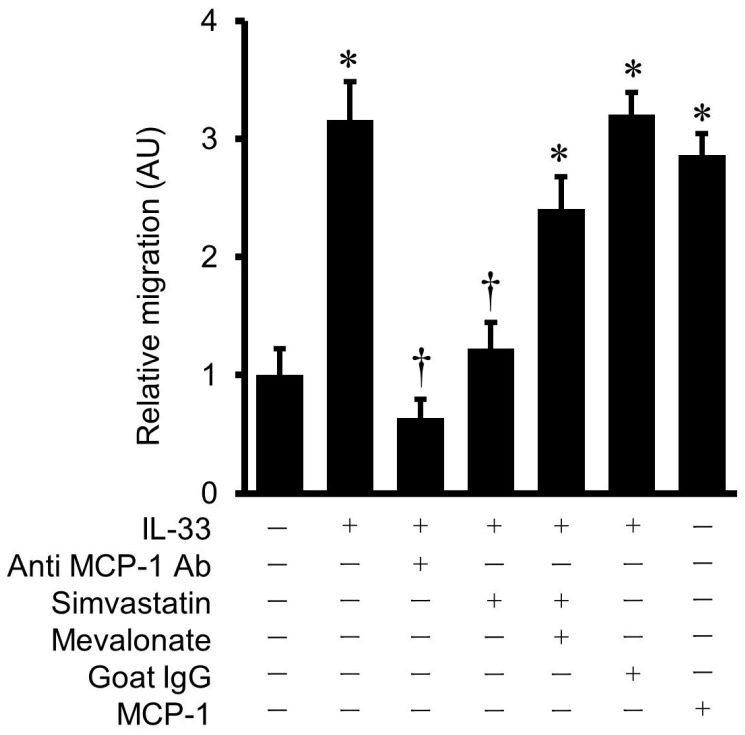
Effects of simvastatin on THP-1 monocyte migration as determined by chemotaxis assay. Relative migration indicates the ratio of migrating THP-1 cells towards the culture medium from HUVECs treated with various reagents relative to those from the untreated cells. THP-1 monocyte chemotaxis was promoted in response to the culture medium treated with 10^−9^ mol/L of IL-33. Pre-incubation of the culture medium with polyclonal anti-MCP-1 antibody (80 μg/mL), but not with goat IgG, resulted in an inhibition of chemotaxis. Simvastatin (10 μmol/L) inhibited IL-33-induced THP-1 monocyte chemotaxis, which was reversed by the addition of mevalonate. Recombinant human MCP-1 (100 nmol/L) served as a positive control. Bars represent mean ± SD of three independent experiments. * *p* < 0.05 vs. untreated control. † *p* < 0.05 vs. IL-33.

**Table 1 ijms-24-13015-t001:** Primers and amplicons of real-time RT-PCR.

Gene Name	Primer Sequences (Forward/Reverse)	Position (nt)	Amplicon Size (bp)
MCP-1	F: 5′-CATAGCAGCCACCTTCATTCC-3′ R: 5′-TCTCCTTGGCCACAATGGTC-3′	109–129 274–293	185
GAPDH	F: 5′-GCACCGTCAAGGCTGAGAAC-3′ R: 5′-TGGTGAAGACGCCAGTGGA-3′	361–380 498–516	138

## Data Availability

The data that support the findings of this study are available from the authors upon reasonable request.

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
