# Peer review of "Inhibitory Effects of Simvastatin on IL-33-Induced MCP-1 via the Suppression of the JNK Pathway in Human Vascular Endothelial Cells"

_ijms, 2023, doi:10.3390/ijms241613015_

Round 1
Reviewer 1 Report (Previous Reviewer 1)
Revisions by the authors are acknowledged.
Author Response
Please see the attachment.

Reviewer 2 Report (New Reviewer)
Comments to author
Manuscript ID - ijms-2574612
Manuscript title: Inhibitory Effects of Simvastatin on IL-33-Induced MCP-1 via Suppressing JNK Pathway in Human Vascular Endothelial Cells
In this manuscript, the author investigates the effects of simvastatin on IL-33-induced MCP-1 expression in human umbilical vein endothelial cells (HU-VECs). IL-33 increased MCP-1 protein expression, while simvastatin inhibited IL-33-induced MCP-1 mRNA, protein secretion, and phosphorylation of JNK and c-Jun. The study suggests that simvastatin may inhibit inflammation progression by selectively suppressing JNK to prevent MCP-1 production.
Minor correction requires
In lines 21, 367, SP600125, and SB203580 what does it mean? Please elaborate.
In line 368, PD98059, and SB203580 what does it mean? Please elaborate.
In lines 412-415, primers (e.g., annealing temp., base pair, and reference) details should be inserted in tabulated form.
In lines 90, 100, 124, 160, 170, 186, and 456 “P” should be italic.
The MS in its present form may be considered for publication in this journal. The authors are suggested to incorporate the minor changes as suggested.
I suggest the quality of this manuscript is good for publication in the International Journal of Molecular Science. but the authors need to pay attention to some minor grammatical errors as well as spelling errors and consistency throughout the text.
Author Response
Please see the attachment.

This manuscript is a resubmission of an earlier submission. The following is a list of the peer review reports and author responses from that submission.
Round 1
Reviewer 1 Report
In Results: 2.1 – Fig. 1 A: It is recommended that the investigators/authors quantify the intensities of immunoreactivity for MCP-1 and provide a graphical plot.
In Results: 2.3 – Fig. 3A-C: It is recommended that the investigators/authors quantify the relative expressions of the phospho-Proteins to their total-Proteins. Also, besides normalizing the phospho-Proteins to their total-Proteins, they should be further analyzed with their blotted housekeeping protein (loading control). The required analysis should be done; showing scanned blots without graphical plots of the analysis outcomes is insufficient.
In Results: 2.7 – Fig. 7A-D: Similar to recommended analysis for data in Fig.3, the relative expressions of the phospho-Proteins to their total-Proteins should be further analyzed by their blotted housekeeping protein (loading control). Additionally, present the representative blot of the housekeeping protein for A-D.
In Results: 2.8 – Fig. 8: The representative images for phospho-c-JUN have too much exposure. Additionally, quantify the expression and present a graphical plot.
Reviewer 2 Report
In this study, the authors showed that Simvastatin inhibited the IL-33 induced expression of MCP-1 protein via JNK pathway. This is an important study in light of the therapeutic strategies for targeting IL-33 associated diseases.
The study is very well designed, exhaustive, and with very interesting data. This manuscript is exceptionally well written and can be accepted in its present form with a small change in the title as follows-
“Inhibitory Effects of Simvastatin on IL-33-Induced MCP-1 via Suppressing JNK pathway in Human Vascular Endothelial 3 Cells”
